# Proteomic Analysis of the Predatory Venom of *Conus striatus* Reveals Novel and Population-Specific κA-Conotoxin SIVC

**DOI:** 10.3390/toxins14110799

**Published:** 2022-11-17

**Authors:** Fabrice Saintmont, Guillaume Cazals, Claudia Bich, Sebastien Dutertre

**Affiliations:** IBMM, Université Montpellier, CNRS, ENSCM, 34093 Montpellier, France

**Keywords:** *Conus striatus*, conotoxin, glycosylation, mass spectrometry

## Abstract

Animal venoms are a rich source of pharmacological compounds with ecological and evolutionary significance, as well as with therapeutic and biotechnological potentials. Among the most promising venomous animals, cone snails produce potent neurotoxic venom to facilitate prey capture and defend against aggressors. *Conus striatus*, one of the largest piscivorous species, is widely distributed, from east African coasts to remote Polynesian Islands. In this study, we investigated potential intraspecific differences in venom composition between distinct geographical populations from Mayotte Island (Indian Ocean) and Australia (Pacific Ocean). Significant variations were noted among the most abundant components, namely the κA-conotoxins, which contain three disulfide bridges and complex glycosylations. The amino acid sequence of a novel κA-conotoxin SIVC, including its N-terminal acetylated variant, was deciphered using tandem mass spectrometry (MS/MS). In addition, the glycosylation pattern was found to be consisting of two HexNAc and four Hex for the Mayotte population, which diverge from the previously characterized two HexNAc and three Hex combinations for this species, collected elsewhere. Whereas the biological and ecological roles of these modifications remain to be investigated, population-specific glycosylation patterns provide, for the first time, a new level of intraspecific variations in cone snail venoms.

## 1. Introduction

Cones are predatory marine molluscs that produce potent venom to subdue prey and deter predators [1]. These venoms are a mixture of a wide variety of conotoxins, which are small peptides (9 to >50 amino acids) often constrained by disulfide bridges [2]. Due to their diversity and specificity, conotoxins are of particular ecological and evolutionary significance and show great promise as drug candidates [3]. Indeed, they often target, with high affinity and high selectivity, a myriad of membrane proteins, including ion channels, GPCRs and transporters [4]. One such conopeptide is a marketed analgesic drug (Ziconotide) and many others have been investigated for the treatment of various disorders, such as schizophrenia, epilepsy, neuromuscular and cardiovascular disorders [5]. The structural diversity of these conopeptides has enabled the development of specialized databases and bioinformatics tools, such as ConoServer, in which the sequences of genes and peptides can be categorized and analyzed [6].

Cone snails have developed various prey-capture methods, depending on their diet. The striated cone, *Conus striatus*, is one of the largest species of fish hunters in the genus *Conus*, which is found throughout the Indo-Pacific region from the Hawaiian Islands to the East African coast. To catch their prey, the piscivorous Pionoconus such as *C. striatus* project an extensible proboscis at the end of which is located the radula, a sharp harpoon-like tooth filled with venom (Figure 1). The injection of venom induces characteristic violent muscle spasms in fish prey, leading to immediate paralysis. This is believed to be achieved by the concerted action of toxins that target the ion channels and receptors in the peripheral nervous system, although the exact mode of action remains controversial. Indeed, a combination of potassium channel blockers and sodium channel activators were originally suggested to explain the observed spastic paralysis [7]. However, the injection of such a combination of conotoxins in the piscivorous cones of the Pionoconus clade seem to be rather unusual, if they occur at all. Rather than being a result of the combination of distinct potassium blockers and sodium channel activators, the spastic paralysis induced by the venom of *C. striatus* is due solely to the presence of the neuroexcitatory κA-conotoxins. Indeed, when κA-conotoxins SIVA and SIVB were purified and injected in fish, these conotoxins alone induce tetanic paralysis, recapping the effect of the whole venom [8,9]. To date, and although being the major toxin used for fish predation, the molecular targets and mechanism of action of these κA-conotoxins have not been elucidated.

Gene-encoded conopeptides are translated from mRNA into protein precursors, which undergo a step-by-step maturation process, during which a signal sequence and a pro-region are cleaved from the mature peptide region. Being amenable to high-throughput sequencing, transcriptomics of the venom gland has revolutionized the field, providing an exhaustive catalogue of all conotoxin sequences in one single experiment [3]. However, in addition to disulfide bonds, the presence of other post-translational modifications (PTMs), such as hydroxylation, N-terminal pyroglutamic acid, C-terminal amidation, bromination or O-glycosylation, can only be detected through mass spectrometry methods [10]. Some of these post-translational modifications may confer structural stability, enzymatic resistance and biological specificity to conotoxins. For instance, SIVA conotoxin from *C. striatus* (ZKSLVPSVITT**CC**GYDOGTM**C**OO**C**R**C**TNS**C-**NH_2_) contains pyroglutamylation (Z), glycosylation of a serine (S), hydroxylation of prolines (O) and C-terminus amidation [8,10]. The determination of the peptide sequence using tandem mass spectrometry, such as collision-induced dissociation (CID) or electron transfer dissociation (ETD), is called de novo sequencing [11]. CID leads to the formation of a, b ions, corresponding to the C-terminus, and y ions corresponding to the N-terminus. The identification of the fragment ions series allows the reconstruction of the peptide sequence and to identify the PTM due to mass differences [11]. Some PTM, such as phosphorylation or glycosylation, are labile and are therefore dissociated before the peptide backbone, preventing the determination of the position of the PTM [12]. However, the fragmentation of the glycosylation allows the determination of its composition and the sugar sequence [13]. On the other hand, ETD form c ions at the C-terminus and z ions at the N-terminus, and do not fragment the labile PTM, allowing the determination of their position. The combination of those two complementary techniques allows the determination of the peptide sequence along with the type and location of PTM [12,14,15].

In this work, we aimed to decipher the intraspecific venom variations between geographically distant specimens of *Pionoconus striatus* from Mayotte Island (Indian Ocean) and Australia (Pacific Ocean). Consequently, we report on the de novo sequencing and glycosylation pattern determination of novel and population-specific κA-conotoxin SIVC, isolated from the predatory venom.

## 2. Results

### 2.1. RP-HPLC of Predatory C. striatus Venoms

For several months, the predatory venom was collected every week from captive specimens in Australia and Mayotte. Freeze-dried samples (5 mg) were separated using RP-HPLC, and 1 min fractions were automatically collected in 2 mL 96 well plates. Both populations produce a comparable and moderately complex venom (Figure 2). Most components eluted between 5 and 55 min, with the dominant peak for each population corresponding to the κA-conotoxins.

### 2.2. Mass Spectrometry Analysis of Venom Fractions

The protonated peptides were observed for the Mayotte fraction 20 at *m*/*z* 1997.8 (2+), 1332.2 (3+) and 999.4 (4+), and for the fraction 21 at *m*/*z* 2018.8 (2+), 1346.2 (3+) and 1009.9 (4+), leading to a monoisotopic neutral mass of 3993.6 Da for the fraction 20, and 4035.6 Da for the fraction 21 (Figure 3). The mass difference of 42 Da between the two fractions could correspond to an acetylation of the peptide. None of the peptide masses correspond to the SIVA conotoxin (4079.6 Da), which is usually found in *C. striatus* [8], but it was observed in the venom extracted from Australian cone snails in the fraction 21 at *m*/*z* 1360.9 (3+) and 1020.9 (4+), leading to a monoisotopic neutral mass of 4079.6 Da (Figure 3, top).

### 2.3. Peptide Glycosylation Investigation

Given that the monoisotopic masses found in the Mayotte venom fractions do not correspond to the known *C. striatus* peptides, the difference may arise from a divergent amino acid sequence and/or glycosylation pattern. Therefore, we first investigated the presence of glycosylation on the peptides from fractions 20 and 21 by CID. As mentioned in the introduction, the fragmentation of the glycosylation occurs before the fragmentation of the peptide, allowing us to determine the composition and the sequence of the sugar moiety. The fragmentation of the 3+ peptide at *m*/*z* 1332 and 1346 for the fractions 20 and 21, respectively, leads to the loss of hexose (Hex) and N-Acetyl Hexose (HexNAc) sugars (see in Figure 4 and in the full spectra in Appendix A). The loss of Hex (162 Da) and HexNAc (203 Da) gives a *m*/*z* difference of 54 and 67.7, respectively, for 3+ fragments and a *m*/*z* difference of 81 and 101.5 for 2+ fragments.

As highlighted in Figure 4, the fragmentation leads to the loss of 4 Hex and 2 HexNAc sugars, and is similar for both fractions, showing that the 42 Da difference is linked to the peptide part and not the glycosylation. The presence of the fragments with low intensity, at *m/z* 1102.5 and 1156.5 for the fraction 20, and at *m/z* 1116.5 and 1170.5 for fraction 21, could correspond to a branch in the glycosylation sequence. The fragments marked with an asterisk correspond to a loss of water (18 Da) from the adjacent fragment ion and can arise from the glycosylation [16] or the peptide [17]. Compared to SIVA, whose glycosylation is Hex_3_HexNAc_2_ (892.3 Da) [8,13], the glycosylation contains one more hexose leading to Hex_4_HexNAc_2_ (1054.4 Da). Therefore, the peptide neutral mass is 2939.30 Da for the fraction 20 and 2981.28 Da for the fraction 21. As the peptide masses do not correspond to any known conotoxin listed in the conoserver [6], the next step towards the identification of the conotoxins is the fragmentation of the peptide to obtain amino acid sequence information.

### 2.4. De Novo Peptide Sequencing

The A-superfamily of conotoxins, to which the SIVA conotoxins belongs to, is characterized by the presence of disulfide bridges with cysteine patterns characteristic of subclasses [18,19]. Therefore, the peptides were reduced using TCEP to obtain information concerning the number of disulfide bridges and to increase the sequence coverage [20].

The comparison between the MS spectra of native and reduced peptides (Appendix A) shows a mass difference of 6 Da, corresponding to three disulfide bonds, as observed in the SIVA conotoxin [8].

The fragmentation of peptide by CID usually leads to a, b and y ions. The a ions are formed by the neutral loss of CO (28 Da) from the b ions, and allows us to differentiate the b and y sequences in the MS spectra (Figure 5 and Appendix A).

As shown in Figure 5 and Appendix A, the a/b ions and y ions sequences were determined for both fractions. The comparison between the two spectra shows that the a and b ions are shifted by 42 Da for the fraction 21, while the y ions have the same mass, indicating that they have the same sequence; however, in the fraction 21, the peptide is acetylated at the N-terminus. The composition of the peptide was deciphered and shown to contain 29 residues, but the order of some amino acids could not be determined due to the lack of intermediate fragments: (acetyl)-AOALxxVVTATTN**CC**GYTG(AO)**C**H(Lxx**C**O)**C**TQT**C**(NH_2_), **C** corresponding to cysteine forming disulfide bridge, Lxx to either leucine or isoleucine and O to 3- or 4-hydroxyproline.

To unambiguously determine the complete sequence, the peptides were digested by α-chymotrypsin, which hydrolyses on the C-terminal side of tyrosine (in position 15) [21,22], leading to two smaller peptides. The peptide AOALxxVVTATTN**CC**GY (Appendix A, *m*/*z* 1277.5 and 1299.5 for the fractions 20 and 21, respectively) is glycosylated, showing that the glycosylation is located either on T7, T9 and/or T10. The fragmentation of the peptide TG(AO)CH(LxxCO)CTQTC(NH_2_) (*m/z* 733.3, Appendix A) allows us to determine the complete peptide sequence of this novel κA-conotoxin, that we named SIVC (Appendix A): (acetyl)-AOALxxVVTATTN**CC**GYTGOA**C**HO**C**Lxx**C**TQT**C**(NH_2_). The cysteine pattern corresponds to the κA-conotoxins family (–CC(X_6–7_)C(X_2_)CXC(X_3_)C–) [18,19].

In Figure 6, we compared the sequence of SIVC with two other conotoxins containing similarities [6]; MIVA from *Conus magus* was found to have both T7 and T9 glycosylated, contributing to a 1053.6 Da higher mass than the predicted peptide [18]. Where the composition of the sugars was not determined, it may be identical to SIVC based on the mass of the sugar moiety. The closest sequence to SIVC was Ac4.2, from *Conus achatinus* (90% identity), the sequence for which was obtained only at the nucleic acid level, and thus lacked PTMs [6,23]. Next, we determined the location of the glycosylation site(s).

### 2.5. Glycosylation Site Determination

As mentioned in the introduction, CID fragments the glycosylation before the peptide, and is thus unable to give information on the glycosylation position. However, ETD fragments the peptide while keeping the glycosylation intact. The results of ETD fragmentation can be found in the supporting information (Appendix A) and show that the glycosylations are located on T7 and T9, each with a glycosylation mass of 527.19 Da that can only correspond to Hex_2_HexNAc. It was not possible to determine the exact sequence for each glycosylation site, but we expect similar sequences as they both have the same composition. Moreover, the sugar linked on the threonine is necessarily a HexNAc [13] and it is not possible to distinguish linear from branched sugar by CID as they present the same fragments. 

## 3. Discussion

The fish-hunting cone snails have certainly evolved one of the most fascinating prey-capture strategies nature has to offer, as it provides one of the rare examples of an invertebrate predator capable of overpowering a vertebrate prey. To achieve this remarkable feat, the piscivorous gastropods from the Pionoconus clade rely on an extremely potent neurotoxic venom that produces immediate immobilization of the prey upon injection [24]. This prey-capture strategy has been nicknamed “hook-and-line” and requires the injection of “lightning-strike cabals”, where a combination of different conotoxin types would produce this electrocution-like effect, hence also the “taser-and-tether” name for this strategy [25]. Originally described in the Atlantic Chelyconus clade, it was hypothesized that two pharmacologically distinct conotoxin classes were necessary to elicit the potent neuroexcitatory effect: a potassium channel blocker and a sodium channel activator [7]. An investigation of the predatory venom of both Chelyconus species *C. ermineus* and *C. purpurascens* later showed dramatic variations in the cocktail of conotoxin injected, with, at times, a complete absence of one or the other “required” conotoxin class, suggesting that “one strategy fits all” does not apply in the case of hook-and-line piscivorous cone snails [26,27]. 

In the Pionoconus clade, *C. striatus* is the largest and most recognizable species. Its venom has been investigated for the past 60 years and many conotoxins have been identified and characterized from its venom duct extracts. These include the α-conotoxins SI, SIA and SII, the μ-conotoxins SIIIA and SIIIB, the δ-conotoxins SVIA-SVIE, con-ikot-ikot and conkunitzin-S1, but also non-disulfide rich peptides such as contryphan-S and conopressin-S. Yet, the major venom component responsible for the immediate spastic paralysis of the fish upon injection are the κA-conotoxins. Indeed, injection of the purified κA-conotoxins SIVA or SIVB recapitulate the observed tetanic effect in fish [8,9]. Quite remarkably, despite being the major ichthyotoxic component, the molecular target remains controversial and its mode of action unknown.

The κA-conotoxins belong to the A-superfamily of conotoxins, together with the shorter α-conotoxins that block the nicotinic acetylcholine receptors. Sequencing of cDNA libraries from several Pionoconus species yielded a large number of κA-conotoxins, and this diversification is consistent with their critical ecological importance [18]. With the exception of SIVA and SIVB, CcTx from *Pionoconus consors* is the best characterized κA-conotoxin. This 30 amino acid-long peptide was found to produce marked contraction and extension of the fins in fish, as well as spontaneous contractions of isolated frog neuromuscular preparations [28]. CcTx possesses the same glycosylation pattern as SIVA and SIVB, and it is the only member of this family for which a complete three-dimensional structure is available, including the sugar moiety [29]. This study determined the exact branching and nature of the sugars, and quite remarkably, identified two terminal L-galactose units, in addition to a D-galactose, a *N*-acetyl-D-galactosamine and a *N*-acetyl-D-glucosamine. The presence of these unexpected L-configured sugars at the terminal extremity explain why it was impossible to remove the glycosylation using conventional D-galactosidases. It also suggests a possible role for the glycosylation in conotoxins, namely the increased stability and resistance to the catabolism of the O-glycan in prey devoid of L-galactosidases.

Here, we describe a novel κA-conotoxin SIVC and its N-acetylated variant from specimens collected in the West Indian Ocean (Mayotte), which differ from previously characterized SIVA and SIVB at the amino acid sequence level and in terms of glycosylation composition. Indeed, in SIVC, the glycosylation was found to consist of Hex_4_HexNAc_2_ (1054.4 Da), as opposed to Hex_3_HexNAc_2_ (892.3 Da) in SIVA, SIVB or CcTx. Furthermore, we determined that the glycosylation is not monosite, as in SIVA (on S7), but rather, is located at two sites (T7 and T9). In this regard, SIVC more closely resembles MIVA from *Pionoconus magus* [18]. The impact of the glycosylation pattern on the structure, stability and biological activity of κA-conotoxins remains to be investigated. Similarly, the role of the N-acetylation is mostly unknown, as it has seldom been studied. For the ω-conotoxin GVIA, N-acetylation was found to significantly (16-fold) reduce the binding affinity [30], whereas it produced only modest changes in affinity (2- to 3-fold) for an α-conotoxin [31].

Quite remarkably, this particular κA-conotoxin SIVC was specific to a population of *Pionoconus striatus* from Mayotte, as it completely replaced the well-known SIVA and SIVB found in the venom of Australian specimens. One could wonder if this could suggest a process of speciation at work, or the presence of a local sub-species (sometimes referred as “*Pionoconus striatus* subsp. *juliaallaryae*”). However, the complete repertoire of conotoxins expressed in the venom gland of Australian specimens of *Pionoconus striatus* was recently revealed using a combination of transcriptomic and proteomic methods [32]. Interestingly, the sequence corresponding to SIVC was found in the transcriptome (named STR18) but seems not to be expressed at the proteomic level (only detected very weakly in the gland extracts of one out of three specimens investigated, but not in the injected venom). This instead suggests that local environment may trigger the expression of some gene and suppress others in response to specific pressures (i.e., different habitats, prey types, predators). Future studies should investigate this hypothesis, including a transcriptomic analysis of specimens from Mayotte to demonstrate the presence (or not) of the SIVA and SIVB conotoxins.

## 4. Conclusions

In this study, we have fully de novo sequenced a novel κA-conotoxin from the predatory venom of *Pionoconus striatus* specimens collected from Mayotte Island. Surprisingly, this κA-conotoxin SIVC was unique to this population (Indian Ocean), and not detected in the predatory venom of Australian specimens. In addition to its novel peptide sequence, the glycosylation was analyzed, and it differs significantly from previously characterized κA-conotoxins from this species. Indeed, the glycosylation was found to consist of 2x(Hex_2_HexNAc) as opposed to Hex_3_HexNAc_2_ identified in SIVA and SIVB. Given that κA-conotoxins of the *Pionoconus* clade are the major venom component injected for prey capture, being responsible for the rapid spastic paralysis of fish, future studies are needed to investigate the role of the glycosylation pattern on the biological activity of these conotoxins.

## 5. Materials and Methods

### 5.1. Materials

Optima LC/MS water and acetonitrile (ACN) were obtained from Fisher Chemical (Illkirch, France). Formic acid (FA), Tris(2-carboxyethyl)phosphine (TCEP), 4-nitrotoluene and α-chymotrypsin from bovine pancreas were obtained from Sigma-Aldrich (Saint-Louis, MO, USA).

### 5.2. Venom Extraction and HPLC Fractionation

Several adult specimens, collected from Mayotte and Australia, were kept in captivity and regularly milked for their predatory venom. Pooled samples from each geographic location were then fractioned by High Performance Liquid Chromatography, operated on an UltiMate^®^ 3000 Standard LC system (ThermoFisher Scientific, Waltham, MA, USA). First, crude venom was loaded onto a C_18_ reversed-phase (RP) HPLC column Luna Omega PS C_18_, 250 mm × 4.6 mm, 5 µm (Phenomenex, Torrance, CA, USA) and separated using a linear gradient water/acetonitrile (ACN) in 0.1% formic acid. Components were eluted at a flow rate of 1 mL/min and 1-min fractions were automatically collected in a 2 mL 96-well plate with the following gradient: 0–80% of ACN over 80 min for a total run of 105 min, at 40 °C. Venom fractions of interest were freeze-dried and resolubilized in 200 µL of pure water. 

### 5.3. Sample Preparation

For mass spectrometry analyses, the κA-conotoxin-containing solutions were diluted 100 times in H_2_O/ACN/FA (90/10/0.1). The reduced samples were obtained by adding TCEP at 0.25 mM final concentration. The solutions were incubated 15 min at 56 °C. The samples were then directly injected.

### 5.4. Digestion

A solution of α-chymotrypsin at 1 mg/mL was prepared in 50 mM acetic acid. The solution was then diluted to 0.01 mg/mL in 50 mM ammonium bicarbonate. The peptides were first reduced by mixing 10 µL of peptide stock solution with 1 µL of 50 mM TCEP, incubated 15 min at 56 °C. After the reduction, 20 µL of α-chymotrypsin 0.01 mg/mL was added and incubated 7 h at 37 °C. The digestion was stopped by adding 1 µL of FA, then diluted by adding 20 µL of ACN and 148 µL of water.

### 5.5. Mass Spectrometry

Electrospray ionization mass spectrometry experiments were performed on a Waters Synapt G2-S (Waters Coproration, Milford, MA, USA). The solutions were infused at a flow rate of 2 µL/min with an automatic injector (Harvard Apparatus, Holliston, MA, USA). The capillary voltage was set to 2 kV, and the source and desolvation temperature to 80 °C and 120 °C, respectively. Collision-induced dissociation (CID) was performed with Argon as collision gas and electron transfer dissociation (ETD) with para-nitrotoluene as the reagent. CID collision energy was optimized depending on the charge state and the type of fragmentation investigated (around 20 V for the glycosylation and between 40 and 60 V for the peptide). For ETD experiments, the discharge current was set to 50 µA, the makeup gas flow to 90 mL/min and the trap RF offset voltage to 425 V with a refill time was 0.2 s. Trap wave height was set to values between 0.15 and 0.3 to optimize the fragmentation.

## Figures and Tables

**Figure 1 toxins-14-00799-f001:**
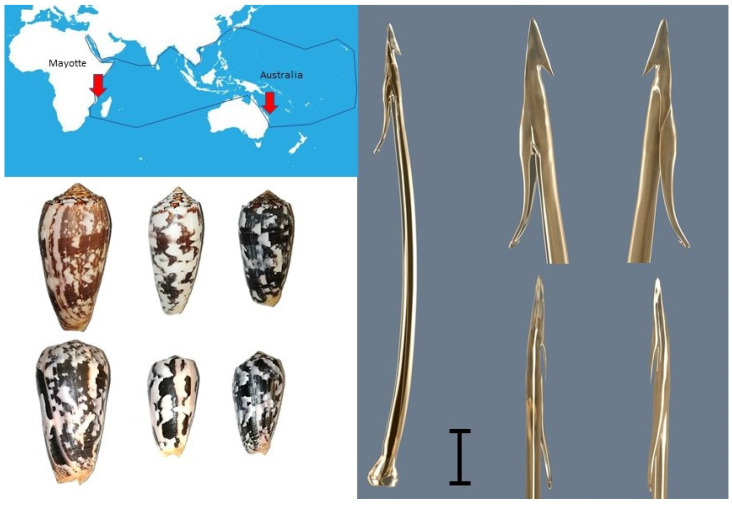
Geographic localisation of the different populations of *Conus striatus* used in this study (**top left panel**). Comparison of the shells of specimens from Australia (**top**) and Mayotte (**bottom left panel**). The spire appears lower in specimens from Mayotte, as well as having a higher shoulder. A 3D rendered model of the radula tooth of *C. striatus* (**right panel**). Vertical bar indicates 1 mm.

**Figure 2 toxins-14-00799-f002:**
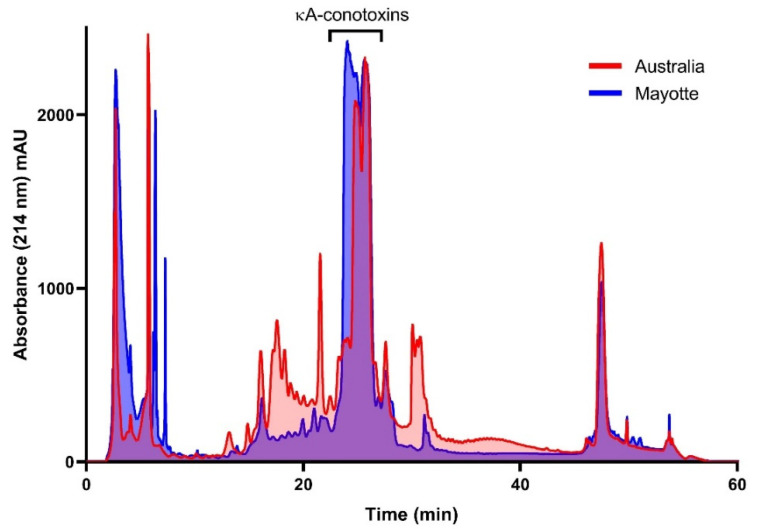
RP-HPLC traces (UV 214 nm) of pooled predatory venom from *C. striatus* collected in Australia and Mayotte. Although they largely overlap, the traces show marked differences, in particular around the κA-conotoxins elution time.

**Figure 3 toxins-14-00799-f003:**
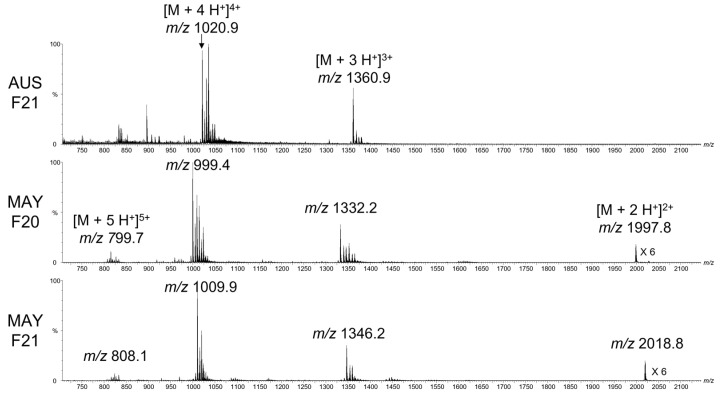
Mass spectra of the fraction 21 from Australia (AUS F21, **top**) and fractions 20 and 21 from Mayotte (MAY F20, **middle** and MAY F21, **bottom**, respectively) of the *Conus striatus* crude venom. The peptides mainly appear protonated with two to four charges, with the monoisotopic *m*/*z* noted on the spectra. The peaks to the right of the protonated ones are sodium and potassium adducts.

**Figure 4 toxins-14-00799-f004:**
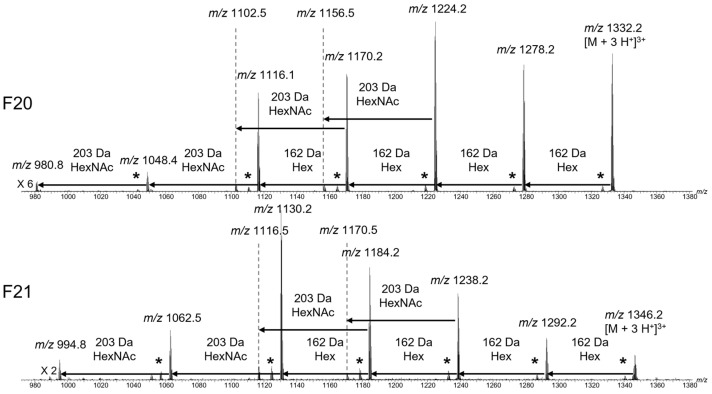
MSMS (CID) spectra of the 3+ charged peptide from the fractions 20 (*m*/*z* 1332, **top**) and 21 (*m*/*z* 1346, **bottom**) from the *C. striatus* crude venom zoomed on the 3+ fragments area. The fragments marked with * correspond to a loss of water (18 Da) from the adjacent fragment ion.

**Figure 5 toxins-14-00799-f005:**
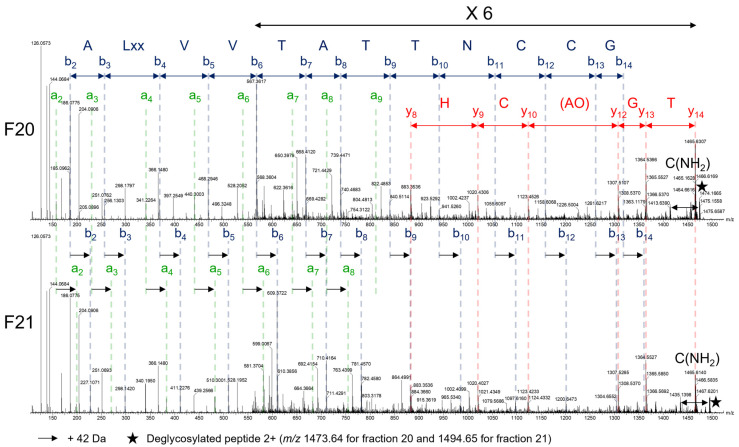
MSMS (CID) spectra of the 3+ reduced peptide from the fractions 20 (*m*/*z* 1334, **top**) and 21 (*m*/*z* 1348, **bottom**) from the *C. striatus* crude venom. The a ions are labelled in green, the b ions in blue and the y ions in red. Lxx corresponds to either leucine or isoleucine, O corresponds to (3 or 4)-hydroxyproline and C(NH_2_) is the C-terminal amidated cysteine. The order of the amino acids in parenthesis was not determined due to the lack of the intermediate fragments. Some b and y ions are not highlighted due to their low intensity.

**Figure 6 toxins-14-00799-f006:**
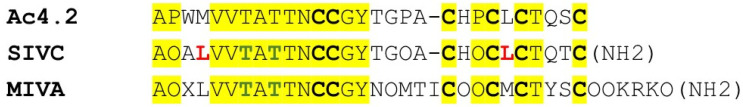
Comparison of the peptide sequence determined in this study (SIVCfrom Mayotte fraction 20) and two other sequences having similarities. Identical amino acids are highlighted in yellow, ambiguous leucines are highlighted in red as they were not confirmed, and glycosylated threonine residues are highlighted in green (see next section for the location of glycosylation sites in SIVC). “X” indicates a gamma-carboxyglutamate.

## Data Availability

Not applicable.

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
