# Peer review of "Proteomic Analysis of the Predatory Venom of *Conus striatus* Reveals Novel and Population-Specific κA-Conotoxin SIVC"

_toxins, 2022, doi:10.3390/toxins14110799_

Round 1
Reviewer 1 Report
In the past, a combination of MALDI-TOF-MS, LSI-MS, and ESI-MS determined that the major O-glycoform of conotoxin SIV A corresponded to a Hex(n)-HexNAc sequence, where HexNAc is linked to a Threonine. Additionally, three less abundant glycosylated forms were observed, i.e., SO4(HexHexNAc), Hex3, and Hex2HexNAc2.
Hence the conotoxins were though to harbor as the major O-glycoform the core-1 type structure, β-d-Galp-(1→3)-α-d-GalpNAc-(1→O)-, that is the T-antigen, one of the most common eukaryotic O-glycan structures.
Here the authors have clearly demonstrated by using a combination of CID and ETS that instead the T-antigen structure is absent in the novel kappa-A-conotoxin SIV C of Conus striatus from Mayotte Island (Indian Ocean) that instead is present in the kappa-A-conotoxin SIV A from Australia (Pacific Ocean).
The O-type glycosylation is clearly shown in figure 4 where the CID showed a fragmentation that has led to the loss of 4 Hex and 2 HexNAc: hence the O-glycosylation of kappa-A-conotoxin SIV C is Hex4HexNAc2.
This could have implication in the predatory range of this species in the Indian Ocean, since the O-glycosylation might play a role in the different biological action of the toxin.
However, this novel O-type glycosylation of kappa-A-conotoxin SIV C is more expected to happen instead in the Asparagine N-type glycosylation sharing the same Hex4HexNAc2 initial structure. But, since in Figure 5 they have determined the kappa-A-conotoxin SIV C peptide structure the only Asn is not having the NXS/T consensus for N-type glycosylation.
Therefore, for how atypical this finding could seem it looks indeed a step ahead in the O-glycosylation of marine derived glycopeptide drugs. It would be interessant to see in future work whether the biological activity is changed respect the other O-glycoform of kappa-A-conotoxin SIV (n).
Author Response
In the past, a combination of MALDI-TOF-MS, LSI-MS, and ESI-MS determined that the major O-glycoform of conotoxin SIV A corresponded to a Hex(n)-HexNAc sequence, where HexNAc is linked to a Threonine. Additionally, three less abundant glycosylated forms were observed, i.e., SO4(HexHexNAc), Hex3, and Hex2HexNAc2.
Hence the conotoxins were though to harbor as the major O-glycoform the core-1 type structure, β-d-Galp-(1→3)-α-d-GalpNAc-(1→O)-, that is the T-antigen, one of the most common eukaryotic O-glycan structures.
Here the authors have clearly demonstrated by using a combination of CID and ETS that instead the T-antigen structure is absent in the novel kappa-A-conotoxin SIV C of Conus striatus from Mayotte Island (Indian Ocean) that instead is present in the kappa-A-conotoxin SIV A from Australia (Pacific Ocean).
The O-type glycosylation is clearly shown in figure 4 where the CID showed a fragmentation that has led to the loss of 4 Hex and 2 HexNAc: hence the O-glycosylation of kappa-A-conotoxin SIV C is Hex4HexNAc2.
This could have implication in the predatory range of this species in the Indian Ocean, since the O-glycosylation might play a role in the different biological action of the toxin.
However, this novel O-type glycosylation of kappa-A-conotoxin SIV C is more expected to happen instead in the Asparagine N-type glycosylation sharing the same Hex4HexNAc2 initial structure. But, since in Figure 5 they have determined the kappa-A-conotoxin SIV C peptide structure the only Asn is not having the NXS/T consensus for N-type glycosylation.
Therefore, for how atypical this finding could seem it looks indeed a step ahead in the O-glycosylation of marine derived glycopeptide drugs. It would be interessant to see in future work whether the biological activity is changed respect the other O-glycoform of kappa-A-conotoxin SIV (n).
The authors wish to thank the reviewer for his insightful comments and appreciation of our work. We fully agree with the last statement and have added in the conclusions "future studies are needed to investigate the role of the glycosylation pattern on the biological activity of these conotoxins."
Reviewer 2 Report
In the manuscript the proteomic analysis of the κA-conotoxin from predatory venom of Conus striatus has been described. The investigation on new, especially, biologically active compounds is very important scientific aspects. The described investigations may contribute to the understanding of the mechanism of action of the Conus striatus venom.
However, the manuscript has some shortcomings.
There is a lack of a clearly formulated research objective.
The conclusions resulting from the described research are also laconic.
For that the advantages and novelty of the described investigations should clearly emphasize especially compared to those previously published information on the subject.
Author Response
In the manuscript the proteomic analysis of the κA-conotoxin from predatory venom of Conus striatus has been described. The investigation on new, especially, biologically active compounds is very important scientific aspects. The described investigations may contribute to the understanding of the mechanism of action of the Conus striatus venom.
However, the manuscript has some shortcomings.
There is a lack of a clearly formulated research objective.
To further emphasize on the objectives of our study, we have added a sentence at the end of the introduction section "In this work, we aimed to decipher the intraspecific venom variations between geographically distant specimens of Pionoconus striatus from Mayotte Island (Indian Ocean) and Australia (Pacific Ocean). "
The conclusions resulting from the described research are also laconic.
We have now added a complete Conclusions section "
- Conclusions
In this study, we have fully de novo sequenced a novel kA-conotoxin from the predatory venom of Pionoconus striatus specimens collected from Mayotte Island. Surprisingly, this kA-conotoxin SIVC was unique to this population (Indian Ocean), and not detected in the predatory venom of Australian specimens. In addition to its novel peptide sequence, the glycosylation was analyzed and it differs significantly from previously characterized kA-conotoxins from this species. Indeed, the glycosylation was found to consist of 2x(Hex2HexNAc) as opposed to Hex3HexNAc2 identified in SIVA and SIVB. Given that kA-conotoxins of the Pionoconus clade are the major venom component injected for prey capture, being responsible for the rapid spastic paralysis of fish, future studies are needed to investigate the role of the glycosylation pattern on the biological activity of these conotoxins."
For that the advantages and novelty of the described investigations should clearly emphasize especially compared to those previously published information on the subject.
We have extensively compared our results to previsouly published information from the same species (Craig AG, Biochemistry, 1998; Kelley WP, Biochemistry, 2006), as well as from other related species (Santos AD, JBC, 2004; Le Gall F, Eur J Neurosci, 1999).
Overall, the authors wish to thank the reviewer for his comments that helped improve our manuscript.
Reviewer 3 Report
I have carefully examined the manuscript entitled “Proteomic analysis of the predatory venom of Conus striatus reveals novel and population-specific κA-conotoxin SIVC”. In this paper the authors describe the analysis of the different composition of conus venom between different populations from Indian and Pacific Ocean. They found a novel κA-conotoxin in Conus striatus from Mayotte islands, compared to the Australian samples, differing in amino acid sequence and glycosylation site and composition. This particular toxin seems to be specific for conus samples living in the Mayotte location, suggesting an important role of that particular environment on the proteome expression in the conus venom.
Authors performed an accurate comparative analysis and structural characterization by mass spectrometry of the venom peptide content of Conus striatus samples coming from Mayotte islands and Australia. Investigations on the glycosylation pattern and sites were done applying CID and ETD induced fragmentation on the peptides isolated from the venom, and the structure of the new conotoxin was achieved by de novo sequencing of the intact and digested peptide.
The manuscript is written in a good English language, the whole structural analysis has been properly carried out and the results correctly interpreted.
Even if the new toxin is strictly connected, from the structural point of view, with the others already reported in literature, the work could be considered as an interesting contribution to the class of marine conotoxins, and I would encourage authors to shed lights on the genomic relevance of the presented results in future studies.
On this basis, I will consider the above-mentioned manuscript suitable for publication on Toxins.
Author Response
I have carefully examined the manuscript entitled “Proteomic analysis of the predatory venom of Conus striatus reveals novel and population-specific κA-conotoxin SIVC”. In this paper the authors describe the analysis of the different composition of conus venom between different populations from Indian and Pacific Ocean. They found a novel κA-conotoxin in Conus striatus from Mayotte islands, compared to the Australian samples, differing in amino acid sequence and glycosylation site and composition. This particular toxin seems to be specific for conus samples living in the Mayotte location, suggesting an important role of that particular environment on the proteome expression in the conus venom.
Authors performed an accurate comparative analysis and structural characterization by mass spectrometry of the venom peptide content of Conus striatus samples coming from Mayotte islands and Australia. Investigations on the glycosylation pattern and sites were done applying CID and ETD induced fragmentation on the peptides isolated from the venom, and the structure of the new conotoxin was achieved by de novo sequencing of the intact and digested peptide.
The manuscript is written in a good English language, the whole structural analysis has been properly carried out and the results correctly interpreted.
Even if the new toxin is strictly connected, from the structural point of view, with the others already reported in literature, the work could be considered as an interesting contribution to the class of marine conotoxins, and I would encourage authors to shed lights on the genomic relevance of the presented results in future studies.
On this basis, I will consider the above-mentioned manuscript suitable for publication on Toxins.
The authors wish to thank the reviewer for his appreciation of our work.